# Plasmid Curing and Exchange Using a Novel Counter-Selectable Marker Based on Unnatural Amino Acid Incorporation at a Sense Codon

**DOI:** 10.3390/ijms222111482

**Published:** 2021-10-25

**Authors:** Yusuke Kato

**Affiliations:** Institute of Agrobiological Sciences, National Agriculture and Food Research Organization (NARO), Oowashi 1-2, Tsukuba 305-8634, Ibaraki, Japan; kato@affrc.go.jp

**Keywords:** counter-selection, unnatural amino acids, pyrrolysyl tRNA synthetase, genetic code expansion, sence codon reassignment, plasmid curing

## Abstract

A protocol was designed for plasmid curing using a novel counter-selectable marker, named *pylS^ZK^-pylT*, in *Escherichia coli*. The *pylS^ZK^-pylT* marker consists of the archaeal pyrrolysyl-tRNA synthetase (PylRS) and its cognate tRNA (tRNA^pyl^) with modification, and incorporates an unnatural amino acid (Uaa), *N^ε^*-benzyloxycarbonyl-l-lysine (ZK), at a sense codon in ribosomally synthesized proteins, resulting in bacterial growth inhibition or killing. Plasmid curing is performed by exerting toxicity on *pylS^ZK^-pylT* located on the target plasmid, and selecting only proliferative bacteria. All tested bacteria obtained using this protocol had lost the target plasmid (64/64), suggesting that plasmid curing was successful. Next, we attempted to exchange plasmids with the identical replication origin and an antibiotic resistance gene without plasmid curing using a modified protocol, assuming substitution of plasmids complementing genomic essential genes. All randomly selected bacteria after screening had only the substitute plasmid and no target plasmid (25/25), suggesting that plasmid exchange was also accomplished. Counter-selectable markers based on PylRS-tRNA^pyl^, such as *pylS^ZK^-pylT*, may be scalable in application due to their independence from the host genotype, applicability to a wide range of species, and high tunability due to the freedom of choice of target codons and Uaa’s to be incorporated.

## 1. Introduction

Counter-selectable markers are genes that promote the death or severe growth inhibition of host cells under appropriate growth conditions [1]. These markers are useful genetic tools for a variety of applications, including creation of chromosomal mutations [2,3,4,5], selection of recombinant vectors [6,7], and plasmid curing [8,9,10]. Plasmid curing refers to the genetic manipulation of removing a plasmid from a bacterium. This technique is important for various genetic engineering applications. For example, stepwise introduction of plasmids and their removal is often required in genome engineering [11,12]. In addition, plasmids for introducing processed DNA sequences, enzyme genes for strain engineering, and screening/characterization tools are usually removed before the final application of the strain [12,13]. Sophisticated plasmid curing uses preincorporation of a counter-selectable marker into the plasmid, culturing under conditions in which the marker exerts toxicity, and then selection of surviving bacteria that lost the plasmid [1].

Many counter-selectable markers were proposed for use in bacteria and there is evidence for the utility of several frequently used markers, but these markers also have inherent drawbacks. The counter-selectable marker, *rpsL*, encodes the S12 ribosomal protein, which is a target of streptomycin [14,15]. Thus, bacteria harboring *rpsL* are selectively killed in the presence of streptomycin. However, use of *rpsL* depends on the host genotype (i.e., the host strain needs to be resistant to streptomycin), which is a severe limitation. Genotype-dependence problems also occur for several other counter-selectable markers, such as *upp* [16], *thyA* [17], and *tdk* [18].

Toxin genes such as *mazF* [19] and *ccdB* [7] were used as genotype-independent counter-selectable markers. Expression of toxin genes is regulated by inducible promoters, and bacteria carrying the toxin gene are killed by addition of the inducer. These promoters must be very tight because leakage expression harms the host cells and facilitates generation of loss-of-function mutations in the toxin genes [20]. Such tightly-regulated expression systems are available only in limited species.

The *Bacillus subtilis sacB* gene may be the most common genotype-independent counter-selectable marker [5]. Levansucrase, which is encoded by *sacB*, converts sucrose to levan, which accumulates in the periplasmic space and causes bacterial cell death [21]. Bacteria harboring *sacB* are selectively removed in the presence of sucrose, but the *sacB* gene is not effective against all bacteria [22,23]. Both Gram-positive bacteria, including *B. subtilis*, and some Gram-negative bacteria are resistant or less sensitive to sucrose selection using *sacB*.

The *pheS** gene is a unique genotype-independent counter-selectable marker [24,25]. This gene is a mutant that has lower substrate specificity than *pheS*, which encodes the α-subunit of phenylalanyl-tRNA synthetase. PheS* incorporates the phenylalanine analog, *p*-chlorophenylalanine (*p*-Cl-Phe), into ribosomally synthesized proteins, resulting in cell death. Bacteria carrying *pheS** are eliminated by incubation in the presence of *p*-Cl-Phe. PheS forms a complex with the host β-subunit PheT to form an active synthetase. Although PheS is highly conserved, there is controversy over how evolutionarily distant a species can be to permit PheS* to compete with host PheS and form a complex with PheT to exert toxicity [26,27]. In addition, undesirable homologous recombination with endogenous *pheS* may reduce the efficiency of counter-selection [26,28].

Recently, we have reported a precise and titratable microbial growth controller regulated by the unnatural amino acid, *N^ε^*-benzyloxycarbonyl-l-lysine (ZK) [29]. This growth controller consists of a pair of genes: ZK-specific modified pyrrolysyl-tRNA synthetase (ZKRS, encoded by *pylS^ZK^*) and its cognate tRNA (tRNA^pyl^, encoded by *pylT*) [30]. The anticodon sequence of tRNA^pyl^ is modified to a sense codon that defines a natural proteinous amino acid. The ZK-regulated growth controller incorporates ZK into ribosomally synthesized proteins at the sense codon, presumably resulting in generation of nonfunctional or toxic abnormal proteins. Consequently, bacteria with growth regulators have inhibited growth or are killed in the presence of ZK, which is nontoxic for normal organisms. The efficacy of growth inhibition and killing strongly depends on the codon selected and the ZK concentration. The ZK-regulated growth controller can suppress the target bacterial growth to almost zero without killing bacteria by choosing the appropriate anticodon sequence and ZK concentration.

Here, we use the ZK-regulated growth controller as a novel, genotype-independent counter-selectable marker under conditions that result in strong growth inhibition or sterilization. The marker is referred to as *pylS^ZK^-pylT*, based on the gene names. The size of the coding region of *pylS^ZK^-pylT* is 1.5 kbp, which is not largely different from established marker genes such as *sacB* (1.4 kb) and *pheS**** (1.0 kb), so *pylS^ZK^-pylT* seems to be within the appropriate size range for a counter-selectable marker. The mechanism of action of *pylS^ZK^-pylT* is similar to that of *pheS**, but *pylS^ZK^-pylT* is expected to have some advantages. Firstly, *pylS^ZK^-pylT* was generated by modifying genes that are unique to archaea, so genes with very high homology do not exist in bacteria. This suggests that undesirable homologous recombination with endogenous genes, as seen with *p**heS**, is unlikely to occur. Next, a wide range of target sense codons can be selected, whereas the target of *pheS** is fixed to the codon assigned to phenylalanine. The anticodon sequence of tRNA^pyl^ is CUA, but this can be modified to other sequences because the pyrrolysyl-tRNA synthetase (PylRS) does not use the anticodon sequence of tRNA^pyl^ for substrate tRNA recognition [31,32]. Growth inhibition by ZK incorporation is highly dependent on selection of the target codon, partially due to the tendency for substitution with physicochemically more distant amino acids to be more toxic [29]. Thus, the freedom to select the target codon is advantageous for maximizing toxicity. Thirdly, the PylRS-tRNA^pyl^ pair works as an orthogonal translation system in both prokaryotes and eukaryotes [33]. In addition, genetic incorporation of many unnatural amino acids, other than ZK, was reported using the PylRS-tRNA^pyl^-based system [31], which suggests that *pylS^ZK^-pylT* and similar systems may be used to perform counter-selection by introducing various unnatural amino acids in a wide range of eukaryotes and prokaryotes. In this study, we show the utility of *pylS^ZK^-pylT* as a counter-selectable marker by establishing a protocol for plasmid curing in *E*. *coli*.

## 2. Results and Discussion

### 2.1. Plasmid Curing

The procedure for plasmid curing using *pylS^ZK^-pylT* is shown in Figure 1a. A typical example of the results is provided by a target plasmid containing the *pylS^ZK^-pylT_GCC_* counter-selectable marker with tRNA^pyl^_GCC_, which has a strong growth inhibitory effect [29]. The sequence of target plasmid is shown in Appendix A. This plasmid also carries the chloramphenicol resistance gene (*cat*). Firstly, bacteria containing the target plasmid were cultured overnight (roughly 10 generations, approximately 18 h) in chloramphenicol-free medium. In this culture, bacteria that do not express *cat* can also survive, so bacteria that have lost the target plasmid will be mixed with other bacteria. Preliminary tests showed that bacteria carrying the target plasmid cannot grow on a solid medium containing ZK due to the effect of *pylS^ZK^-pylT_GCC_*. The frequency of ZK-resistant and chloramphenicol-resistant escapers was estimated to be <70 ppm (0 escapers/1.4 × 10^4^ total inoculation). In contrast, bacteria without the target plasmid grew normally and formed colonies. The bacterial suspension after cultivation was diluted, and then seeded on the solid medium. Approximately 2% of the inoculated bacteria, which was much higher than the expected frequency of escapers, were isolated as ZK-resistant colonies that had lost the plasmid (Figure 1b). The loss of the plasmid results in ZK resistance and chloramphenicol sensitivity due to loss of *cat*. All 64 colonies randomly picked up from the isolated ZK-resistant colonies were chloramphenicol-sensitive, confirming loss of the plasmid (Figure 1c). These results suggest that *pylS^ZK^-pylT_GCC_* can be used as a counter-selectable marker for reliable plasmid curing.

### 2.2. Plasmid Exchange

Maintaining plasmids by using antibiotic resistance genes as selection markers is not desirable on an industrial scale because of the high cost of antibiotics and their negative impact on the environment [34]. There are also situations in which antibiotics cannot coexist, such as in vaccines for animals and humans [35,36]. An alternative method of maintaining plasmids without selection markers is to delete an essential gene on the genome and to locate a gene on the plasmid that complements the function of the gene [37]. If the plasmid is to be exchanged for another similar plasmid (e.g., a series of partially modified plasmids), simple introduction of a new plasmid after plasmid curing is not possible because the cell cannot survive without maintaining a plasmid carrying the complementary gene. Therefore, we developed a protocol to exchange similar plasmids with the same replication origin and antibiotic resistance genes without plasmid curing (Figure 2a).

The target plasmid contains the counter-selection marker, *pylS^ZK^-pylT_GGU_+pylT_CGC_*, including two tRNA^pyl^ genes, tRNA^pyl^_GGU,_ and tRNA^pyl^_CGC_, both of which show a strong growth inhibitory effect [29]. We attempted to exchange the target plasmid directly with the parent plasmid, pACYC184 [38]. The sequences of target plasmid and pACYC184 are shown in Appendix A, respectively. Only pACYC184 has another selection marker (the tetracycline-resistance gene), but both the target plasmid and pACYC184 contain *cat.* In this experiment, the phenotype of tetracycline resistance was used only to detect bacteria with pACYC184 and not as part of the plasmid exchange procedure. A typical example of the results is as follows. pACYC184 was first introduced into bacteria containing the target plasmid by electroporation. The ratio of tetracycline-resistant bacteria carrying pACYC184 was determined just after electroporation (Figure 2b). At this stage, a few bacteria carried pACYC184 (3 × 10^−4^) and most bacteria had the target plasmid only.

The bacterial mixture was diluted and grown in liquid culture in the presence of chloramphenicol. In this process, the target plasmid is stochastically dropped out and bacteria carrying only pACYC184 are generated. Subsequently, to amplify bacteria that only harbor pACYC184 selectively, we diluted and amplified in liquid medium containing ZK and chloramphenicol. Finally, the rediluted cultures were inoculated on solid medium containing ZK and chloramphenicol to isolate colonies in which the target plasmid was replaced by pACYC184 (Figure 2c). In this solid selection medium, the escaper frequency of bacteria with the target plasmid (i.e., chloramphenicol- and ZK-resistant) was <1 ppm (0 escapers/9.4 × 10^5^ total inoculation).

Loss of the target plasmid and acquisition of pACYC184 were confirmed in two ways. Firstly, all randomly isolated ZK-resistant colonies were tetracycline-resistant (24/24), suggesting that all of these colonies acquired pACYC184 (Figure 2c). Secondly, the maintained plasmids were identified by PCR using primer sets specific for the target plasmid and pACYC184, respectively (Figure 2d, Appendix A). In all ZK-resistant colonies, only pACYC184 was detected, with no target plasmid. This suggests that the isolated colonies were only those in which the target plasmid was completely replaced by pACYC184. These results show that *pylS^ZK^ − pylT_GGU_ + pylT_CGC_* is useful as a counter-selectable marker for plasmid exchange between homologous plasmids.

In conclusion, *pylS^ZK^* − *pylT* functioned well as a counter-selectable marker that can be used for plasmid curing and homologous plasmid exchange in *E*. *coli*. To apply this counter-selectable marker to other bacterial species, it will be important to express these two genes at the appropriate intensity and dosage ratio to allow ZK to be introduced efficiently [39]. There are no anticipated major technical barriers for expression of these genes, as the use of constitutive promoters is sufficient. In addition, the target sense codon with the highest screening efficiency has to be selected [29]. Since the optimal target sense codon is affected by many factors, it is difficult to predict theoretically and will need to be determined experimentally for each species.

## 3. Materials and Methods

### 3.1. Bacterial Strain, Culture, and Plasmid Transfection

BL21-AI [*F ompT gal dcm lon hsdSB (rB mB) araB*:*T7RNAP tetA*] was used throughout the study [40]. Bacteria were grown in Luria-Bertani (LB) medium at 37 °C. Chloramphenicol (50 μg/mL, final) and/or tetracycline (15 μg/mL, final) were added as required. For preparation of solid medium, an aliquot of agar (2%) was added. Plasmid transfection was carried out by electroporation using a Gene Pulser II electroporator (BIO-RAD).

### 3.2. Plasmid Curing

The procedure for plasmid curing is illustrated in Figure 1a. We previously constructed a target plasmid carrying *pylS^ZK^* and *pylT_GCC_* [29], which causes strong growth inhibition in the presence of ZK, from *pTK2-1 ZLysRS1* supplied by the Yokoyama and Sakamoto laboratory (RIKEN, Japan) [30]. To generate bacteria that lost the target plasmid, bacteria carrying this plasmid were cultured overnight (about 10 generations) in a medium that did not contain the chloramphenicol selection marker. Bacterial cultures were diluted 10,000-fold in fresh medium, and 250 µL was inoculated onto antibiotic-free solid medium with and without 3 mM ZK. Colonies that grew in medium containing ZK were isolated as bacteria that had lost the target plasmid. Loss of the plasmid was confirmed by chloramphenicol sensitivity due to the loss of the chloramphenicol-resistance gene located on the target plasmid.

### 3.3. Plasmid Exchange

The procedure of plasmid exchange is illustrated in Figure 2a. We prepared an *E*. *coli* strain with the target plasmid containing *pylS^ZK^* and its two cognate tRNA genes: *pylT_GGU_* and *pylT_CGC_*, both of which exhibit strong growth inhibition in the presence of ZK [29]. This strain was further transfected with pACYC184, the parent plasmid of the target plasmid. The target and parent plasmids are similar plasmids that contain *cat* and the *p15A* replication origin. After transfection, the cells were allowed to recover for 2 h at 37 °C in SOC medium. After 1000-fold dilution, the cells were incubated overnight at 37 °C in LB medium containing chloramphenicol to generate bacteria without the target plasmid. Then, to kill bacteria that maintained the target plasmid and selectively grow only those with loss of this plasmid, the cells were diluted 100-fold (10 μL/1 mL) in LB medium containing 3 mM ZK and chloramphenicol, and further incubated at 37 °C for 8 h. After selection in the liquid medium, the culture was diluted 10^−6^ times and 250 μL of the diluted culture was inoculated in solid medium containing 3 mM ZK and chloramphenicol. Then, 24 of the resulting colonies were randomly picked up for tests to identify plasmids to be maintained. Maintenance of pACYC184 was determined by tetracycline resistance due to the *tet* gene, which is present only in this plasmid and not in the target plasmid. In addition, the harbored plasmids were identified by PCR specific for the target plasmid or pACYC184.

## Figures and Tables

**Figure 1 ijms-22-11482-f001:**
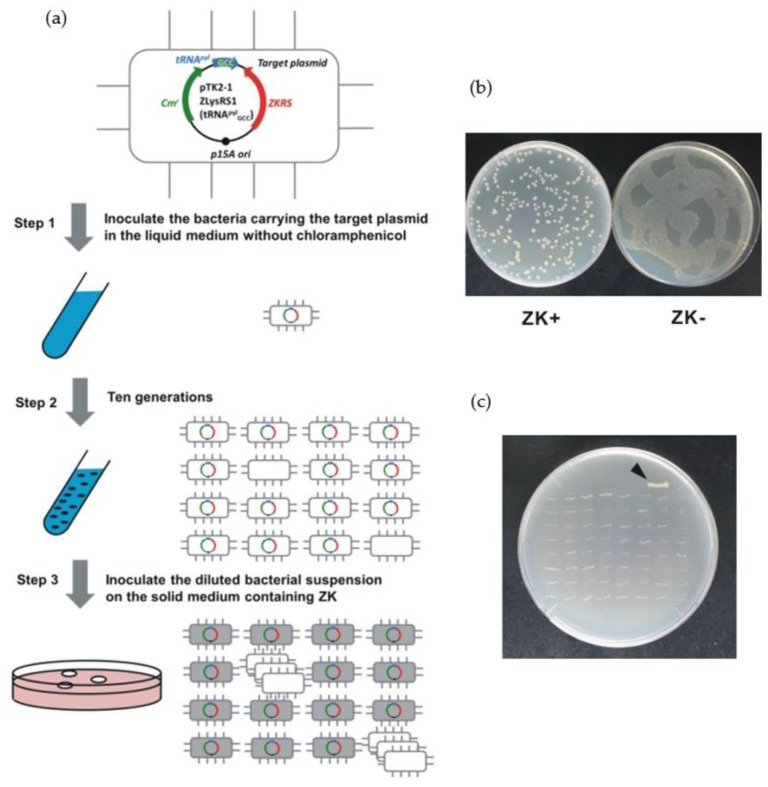
Plasmid curing. (**a**) Schematic diagram of procedure. For detailed experimental conditions, see Materials and Methods. (**b**) Selection of bacteria that lost target plasmid in Step 3. Bacterial culture obtained in Step 2 was diluted and same volume was inoculated as several drops onto solid medium with or without 3 mM ZK. Then, drops were spread by tilting plates. Resulting “twiggy pattern” of fused colonies was an artifact of this incomplete coverage of plate. (**c**) Confirmation of target plasmid loss. Sixty–four colonies obtained in Step 3 were inoculated on solid medium containing chloramphenicol. Parent strain (black arrowhead) carrying target plasmid proliferated, but all 64 of isolates were chloramphenicol–sensitive.

**Figure 2 ijms-22-11482-f002:**
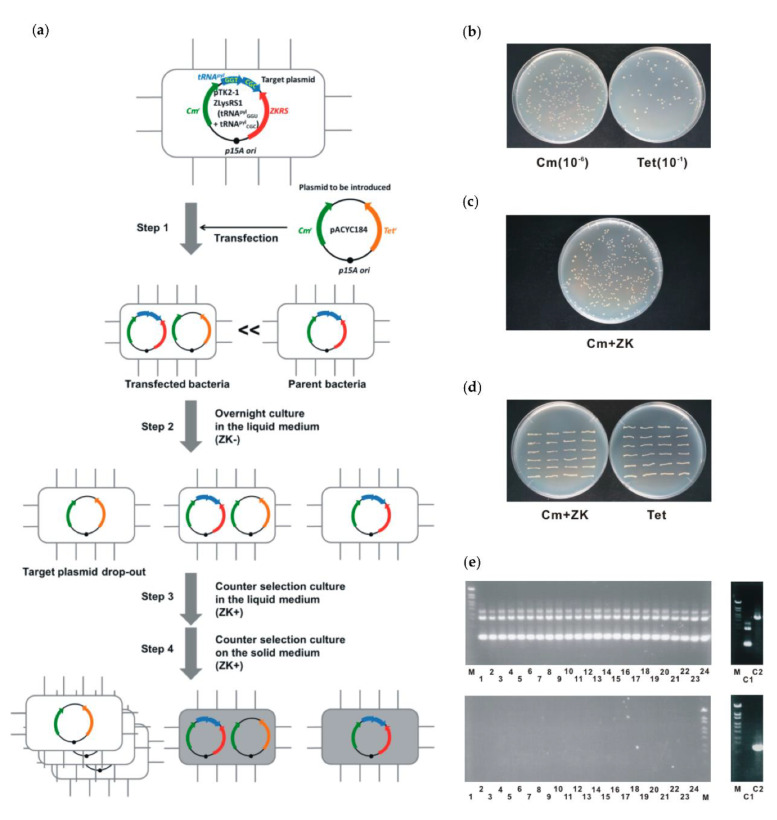
Plasmid exchange. (**a**) Schematic diagram of procedure. For detailed experimental conditions, see Materials and Methods. (**b**) Transfection of substitute plasmid. From bacterial suspension immediately after transfection in Step 1, bacteria carrying substitute plasmid were detected using tetracycline resistance. (**c**) Isolation of bacteria in which target plasmid was lost and only substitute plasmid was maintained. In Steps 3 and 4, bacteria were inoculated onto solid medium containing 3 mM ZK to eliminate those with target plasmid. Since medium also contains chloramphenicol, only bacteria that carried substitute plasmid grew. (**d**) Confirmation of maintenance of the substitute plasmid. Twenty–four colonies picked up randomly in Step 4 were tested for sensitivity to tetracycline. All tested colonies were tetracycline–resistant, suggesting that they all carried the substitute plasmid. (**e**) Specific detection of target and substitute plasmid by PCR. For 24 colonies, substitute plasmid (top row) and target plasmid (bottom row) were detected by PCR using specific primer sets. Bacteria carrying substitute plasmid (C1) and target plasmid (C2) were used as controls. M, molecular mass marker (λ/StyI digest).

## Data Availability

The data presented in this study are available on request from the corresponding author.

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
