# Peer review of "Plasmid Curing and Exchange Using a Novel Counter-Selectable Marker Based on Unnatural Amino Acid Incorporation at a Sense Codon"

_ijms, 2021, doi:10.3390/ijms222111482_

Round 1

Reviewer 1 Report

In this study, the author reported the development of a novel counter-selectable marker for plasmid curing and exchange in bacteria. This new marker is based on an unnatural amino acid (UAA) incorporation system, which consists of an archaeal pyrrolysyl-tRNA synthetase and a modified version of its cognate tRNA. When the counter-selectable marker is expressed in bacteria, the addition of UAA in the growth medium results in the incorporation of UAA into the sense sequence of the endogenous bacterial proteins and therefore affects the growth and survival of the host bacteria. Using this marker, the author successfully demonstrated plasmid curing and exchange in E.coli with high fidelity and efficiency. In theory, this novel counter-selectable marker has overcome most of the disadvantages of the currently common methods, such as rpsL and toxin genes under the control of inducible promoters. The reported counter-selectable marker should find applications in a broad spectrum of bacteria species. Overall, the current manuscript is in an above-average quality, with the experiments appropriately designed and the results clearly presented. I would suggest accepting the manuscript after the following minor revision is addressed.

Minor points:

  1. The size of the counter-selectable marker should be discussed, in comparison with other existed markers. 

Reviewer 2 Report

In this article the author reports the repurposing of an already described two-gene system as a counter selective marker for plasmid curing and substitution. I believe that the described protocols could be useful for the community. Although very well written I have a few concerns regarding the presented data:

1. Supplementary figures.

1.1. As it is, the text suggests that in the supplementary figures, data evidencing strong inhibitory effects can be found. That is not correct as the supplementary figures correspond solely to plasmid sequences. This should be changed for clarity.

1.2. If the journal rules so permit, I would strongly urge the author to provide the plasmid sequences as genebank file (or other sequence format that allows annotations).

2. Main figures.

2.1. In figure 1b and under the conditions ZK-, it looks like there is growth as a film or pellicle but not as isolated colonies. This aspect should be clarified by the author.

2.1.1. Is that what is happening?

2.1.2. Why is that happening?

2.1.3. What are the implications of this to the success of the protocol?

2.2. In figure 2e there is some incoherence concerning the sizes of the bands that should be clarified.

2.2.1. Can the author annotate the sizes on the image or indicate on the legend which DNA ladder was used?

2.2.2. Can the sequence of the primers be provided in the methods section or annotated in the sequences?

2.2.3. Why is the marker for the control samples provided as a distinct panel?

2.2.4. If that independently supplied marker image belongs to the gel of the control samples why are the sizes of the PCR products smaller on the control gel than on the sample gel?

Author Response

Reviewer 2

In this article the author reports the repurposing of an already described two-gene system as a counter selective marker for plasmid curing and substitution. I believe that the described protocols could be useful for the community. Although very well written I have a few concerns regarding the presented data:

Reply: We thank the reviewer for the constructive comments. The manuscript has been revised in response to these comments, as follows.

  1. Supplementary figures.

1.1. As it is, the text suggests that in the supplementary figures, data evidencing strong inhibitory effects can be found. That is not correct as the supplementary figures correspond solely to plasmid sequences. This should be changed for clarity.

Reply: The description has been revised according to the reviewers' suggestions (line 121-124 and line 154-158 in the revised manuscript).

1.2. If the journal rules so permit, I would strongly urge the author to provide the plasmid sequences as genebank file (or other sequence format that allows annotations).

Reply: The plasmid sequence format has been changed to the format suggested by the reviewer.

  1. Main figures.

2.1. In figure 1b and under the conditions ZK-, it looks like there is growth as a film or pellicle but not as isolated colonies. This aspect should be clarified by the author.

2.1.1. Is that what is happening?

2.1.2. Why is that happening?

2.1.3. What are the implications of this to the success of the protocol?

Reply: In this experiment, the same volume of bacterial culture was inoculated onto ZK+ and ZK- plates. The bacteria carrying the plasmid died on the ZK+ plate, so only a small population survives and forms isolated colonies. In contrast, all bacteria survived and became confluent on the ZK- plate. The reason why the bacteria do not grow evenly in the ZK- plate is because they were inoculated unevenly. After the culture was dripped, the plate was tilted to let the drops flow and spread over the surface, but there were areas where the culture solution did not spread. An explanation has been added to clarify the above (line 112-115).

2.2. In figure 2e there is some incoherence concerning the sizes of the bands that should be clarified.

2.2.1. Can the author annotate the sizes on the image or indicate on the legend which DNA ladder was used?

Reply: The DNA ladder used has been added to the legend (line 179).

2.2.2. Can the sequence of the primers be provided in the methods section or annotated in the sequences?

Reply: The primer sequences were added as annotations to the plasmid sequences shown in Fig S2 and S3 (line 193).

2.2.3. Why is the marker for the control samples provided as a distinct panel?

Reply: With this agarose gel, only a maximum of 25 samples can be analyzed. Therefore, the analysis of the control experiment was done as an independent electrophoresis.

2.2.4. If that independently supplied marker image belongs to the gel of the control samples why are the sizes of the PCR products smaller on the control gel than on the sample gel?

Reply: I agree with the reviewer's point. This is due to the fact that the control sample was analyzed together with a sample unrelated to this study, which resulted in a misalignment of the molecular mass markers due to distance. In the revised manuscript, the control experiment was retried. In the revised Figure 2e, the molecular mass marker and the control sample were analyzed next to each other and presented as a single picture.

Round 2

Reviewer 2 Report

The author has successfully addressed my previous concerns. I am satisfied and hence, recommend that the manuscript is accepted in its current form.